# Associations between Single Nucleotide Polymorphisms from the Genes of Chemokines and the CXCR2 Chemokine Receptor and an Increased Risk of Endometrial Cancer

**DOI:** 10.3390/cancers15225416

**Published:** 2023-11-14

**Authors:** Wioletta Izabela Wujcicka, Agnieszka Zając, Krzysztof Szyłło, Hanna Romanowicz, Beata Smolarz, Grzegorz Stachowiak

**Affiliations:** 1Scientific Laboratory of the Center of Medical Laboratory Diagnostics and Screening, Polish Mother’s Memorial Hospital—Research Institute, 93-338 Lodz, Poland; 2Department of Operative Gynecology and Gynecologic Oncology, Polish Mother’s Memorial Hospital—Research Institute, 93-338 Lodz, Poland; agnieszka.zajac@icloud.com (A.Z.); krzysztof.szyllo@iczmp.edu.pl (K.S.); gstach23@interia.pl (G.S.); 3Department of Operative and Endoscopic Gynecology, Medical University of Lodz, 93-338 Lodz, Poland; 4Department of Clinical Pathomorphology, Polish Mother’s Memorial Hospital—Research Institute, 93-338 Lodz, Poland; hanna.romanowicz@iczmp.edu.pl; 5Laboratory of Cancer Genetics of the Department of Clinical Pathomorphology, Polish Mother’s Memorial Hospital—Research Institute, 93-338 Lodz, Poland; beata.smolarz@iczmp.edu.pl

**Keywords:** endometrial cancer, chemokines, chemokine receptor, single nucleotide polymorphisms (SNPs), *CCL2*, *CCL5*, *CXCL8 (IL8)*, *CXCR2*

## Abstract

**Simple Summary:**

Endometrial cancer is the second most common tumor of the female reproductive organs in the world. Taking into account the immunological mechanisms of defense against cancer, an important role is attributed to cytokines. In this case-control study, we aimed to identify the possible associations between selected single nucleotide polymorphisms (SNPs), localized in the *CCL2*, *CCL5*, *CXCL8*, and *CXCR2* genes, and the onset and progression of endometrial cancer. We found that *CCL5* and *CXCR2* polymorphisms were associated with increased cancer risk, while the relationships remained significant after adjustments for age, diabetes, hypertension, or endometrial thickening. The selected haplotypes for *CCL5* and *CCL2* SNPs also correlated with an increased risk of cancer. We concluded that the four polymorphisms studied were significantly associated with an increased risk of endometrial cancer. The obtained results may be useful for identifying the signaling pathways involved in observed genetic changes, which is important for the tumorigenesis of endometrial cancer.

**Abstract:**

Significant relationships with endometrial cancer were demonstrated, both for *CCL2*, *CCL5*, and *CXCL8* chemokines and for the chemokine receptor *CXCR2*. The reported case-control study of genetic associations was designed to establish the role of selected single nucleotide polymorphisms (SNPs) of the *CCL2*, *CCL5*, *CXCL8*, and *CXCR2* genes in the onset and progression of endometrial cancer. This study was conducted on 282 women, including 132 (46.8%) patients with endometrial cancer and 150 (53.2%) non-cancerous controls. The genotypes for *CCL2* rs4586, *CCL5* rs2107538 and rs2280789, *CXCL8* rs2227532 and −738 T>A, and *CXCR2* rs1126580 were determined, using PCR-RFLP assays. The AA homozygotes in *CCL5* rs2107538 were associated with more than a quadruple risk of endometrial cancer (*p* ≤ 0.050). The GA heterozygotes in the *CXCR2* SNP were associated with approximately threefold higher cancer risk (*p* ≤ 0.001). That association also remained significant after certain adjustments, carried out for age, diabetes mellitus, arterial hypertension, or endometrial thickness above 5 mm (*p* ≤ 0.050). The A-A haplotypes for the *CCL5* polymorphisms and T-A-A haplotypes for the *CCL2* and *CCL5* SNPs were associated with about a twofold risk of endometrial cancer (*p* ≤ 0.050). In conclusion, *CCL2* rs4586, *CCL5* rs2107538 and rs2280789, and *CXCR2* rs1126580 demonstrated significant associations with an increased risk of endometrial cancer.

## 1. Introduction

Endometrial cancer is a common tumor of the female reproductive system, ranking second in the world after cervical cancer [1,2]. It develops in the endometrium, lining the uterine cavity and smoothly flowing into the mucosa of the cervical canal. Two types of endometrial cancer are known, differing in molecular background, aggressiveness, and the age of onset [3,4,5]. The well-promising type I is more prevalent, occurring in the perimenopausal age, and developing from endometrial hyperplasia, after stimulation with estrogens [3,4]. In turn, type II, associated with a worse prognosis, is generally less frequent and targets women in the sixth and seventh decade of their lives, while not being associated with hormonal stimulation [3]. Most cases of endometrial cancer occur in highly developed countries, where it is the fourth most prevalent malignant tumor in women, after breast, lung, and skin cancer, and the most common genital cancer [6,7,8,9]. From a general perspective, the incidence of endometrial cancer is slowly but steadily increasing [4,10]. Most cases are diagnosed early, when the chances of recovery are still high and the five-year relative survival at diagnosis is estimated at over 80–90% [11]. However, the advanced forms of endometrial cancer are characterized by a worse prognosis and require more aggressive treatment, bringing a cure to approximately 30–50% of affected patients [3,5,11].

Many studies have recently aimed at improving the clinical management of endometrial cancer, personalizing patient therapy, adding novel molecular analyses to define cancer risk classes, and developing therapies, based on carcinogenic molecules [12,13]. Taking into account mutations and somatic copy number variations, genome and exome sequencing, and the microsatellite instability (MSI) test, endometrial cancer can be divided into the following four groups, each one being associated with different prognoses in terms of specific progression-free survival (PFS) and the risk of recurrence: polymerase epsilon (POLE) ultramutated, MSI hypermutated, copy-number (CN) low, and CN high [13,14]. In addition, a new model, called ProMisE (Proactive Molecular Risk Classifier for Endometrial Cancer), based on the Institute of Medicine (IOM) guidelines, has been introduced to overcome the methodology limitations in the Cancer Genome Atlas (TCGA) study, including the costs, the complexity, and the lack of immediate clinical application [13].

Among the immunological mechanisms of defense against cancer, an important role is assigned to the activity of cytokines, belonging to many groups, including: interleukins, interferons, chemokines, and the tumor necrosis factor (TNF) superfamily [4,15,16]. Cancer cells can secrete both monocyte and macrophage chemotactic factors, e.g., C-C motif chemokine ligand 2 (CCL2), as well as the agents that accelerate the maturation of these cells and activate them, e.g., the colony stimulating factor 1 (CSF1) and CSF2, and also inhibit their chemotaxis [4,6,8,17]. Moreover, the chemokines, secreted by macrophages, may have an opposite effect on tumor growth [17]. They can chemotactically attract many cells (neutrophils, monocytes, and effector T lymphocytes) and increase their inflow into the tumor [16,17]. Some of the chemokines, including the platelet factor 4 (PF4), the C-X-C motif chemokine ligand 9 (CXCL9), and CXCL10, may inhibit tumor vessel formation [17]. In turn, other chemokines, including CCL2, CCL5, CXCL1, CXCL5, and CXCL8, can stimulate angiogenesis [8,15,18,19].

Considering endometrial cancer, a significant relationship was noted between its incidence and pathogenesis on the one hand and CCL2, CCL5, and CXCL8 chemokines, and their selected receptors, including CCR2 (the receptor for CCL2) and CXCR2 (the receptor for CXCL8) on the other [3,5,6,8,9]. Genetic studies confirm the association of the single nucleotide polymorphisms (SNPs) −2518 G>A (rs1024611) of the *CCL2* gene and 190 G>A (rs1799864) of the *CCR2* gene with the occurrence of endometrial cancer [6,16]. In this reported case-control genetic association study, we aimed to determine the role of selected SNPs of the *CCL2* (rs4586 (903 T>C)), as well as *CCL5* (rs2107538 (−403 G>A) and rs2280789 (351 A>G)), *CXCL8* (*IL8*, rs2227532 (−845 T>C) and −738 T>A), and *CXCR2* (rs1126580 (1440 G>A)) genes, in the onset and progression of endometrial cancer.

## 2. Materials and Methods

This study involved 282 women, including 132 (46.8%) patients with endometrial cancer and 150 (53.2%) non-cancerous control individuals (see Table 1). The females, classified in the research project, were all hospitalized at the Department of Gynecology and Oncological Gynecology of the Polish Mother’s Memorial Hospital—Research Institute in Lodz, Poland. The women with endometrial cancer were aged between 44 and 88 years, while the controls were aged between 36 and 75 years. Further clinical characteristics of the enrolled women, including diabetes mellitus, arterial hypertension, and endometrial thickness, are presented in Table 1. The endometrial cancers were graded and staged, according to the criteria of the International Federation of Gynecology and Obstetrics (FIGO) [20]. For the non-cancer control group, normal endometrial tissue specimens were analyzed. Both cancerous and non-cancerous endometrial samples were collected by the dilation and curettage (D&C) procedure, performed on medical grounds. All the samples, previously collected for diagnostic purposes, were anonymized in the reported project. This study was approved by the Research Ethics Committee at the Polish Mother’s Memorial Hospital—Research Institute (the Institutional Review Board, approval number 42/2018). Informed consent forms were signed by all the study women, as recommended by the Research Ethics Committee.

### 2.1. DNA Extraction

Genomic DNA of the women with endometrial cancer and of the control individuals was extracted from paraffin-embedded sections, using a Syngen DNA Micro Kit (Cat No. SY244020, Syngen Biotech, Wroclaw, Poland). The obtained DNA was diluted in 100 µL of elution buffer and stored at −20 °C until further genetic studies.

### 2.2. Genotypes within SNPs of Chemokine and Chemokine Receptor Genes

The genotypes from *CCL2* 903 T>C, *CCL5* −403 G>A and 351 A>G, *IL8* −845 T>C and −738 T>A, as well as *CXCR2* +1440 G>A polymorphisms were assayed by the PCR-RFLP method (see Figure 1). Primer sequences, annealing temperatures, and the amplicon lengths, specific to the performed PCR methods, are presented in Table 2. The external oligonucleotides for all the polymorphisms were obtained from the literature [21,22,23,24]. Additional internal primer sequences for nested PCR assays were designed, using the PerlPrimer v1.1.21 software. PCR products were resolved on 1% agarose gels, stained with ethidium bromide, and then digested overnight at 37 °C, with SatI, RsaI, MboII, AseI, XbaI, or HphI restriction enzymes, to estimate genotypes within the analyzed polymorphic sites (see Table 3). Genotypes were determined, based on the restriction profiles of digested products, obtained on 2% agarose gels (see Figure 2 and Appendix A, Table 2 and Appendix A). The selected PCR products were additionally sequenced by the Sanger method at the Genomed Joint-Stock Company (Warsaw, Poland) to corroborate the genotypes previously estimated by the PCR-RFLP method. For *CCL2* SNP, DNA fragments were sequenced for ten TT, six CT, and six CC genotypes. In case of *CCL5* −403 G>A polymorphism, nine GG and 12 GA genotypes were verified by a sequencing process. Regarding *CCL5* 351 A>G SNP, PCR products were sequenced for seven AA and eight AG genotypes. In case of *IL8* −845 T>C SNP, sequencing was performed for 14 TT genotypes, while, in case of *IL8* −738 T>A, sequencing was performed for seven TT and two TA genotypes. The sequenced DNA fragments were analyzed by the Sequence Scanner 1.0 (Applied Biosystems) program.

### 2.3. Statistical Analysis

The characteristics of both the women with endometrial cancer and the non-cancerous controls were compared, using the NCSS 2004 software. The Mann-Whitney U test was performed to compare the ages of the examined groups of women. Differences were estimated in the prevalence of diabetes mellitus, arterial hypertension, and endometrial thickness above 5 mm, and alleles in the tested polymorphisms of the examined women, using Pearson’s Chi-squared test. The distribution of genotypes and alleles was estimated within the analyzed SNPs and haplotypes for *CCL2* and *CCL5* polymorphisms in the studied groups of women by means of descriptive statistics, supported by the SNPStats software [25]. Hardy-Weinberg (H-W) equilibrium was analyzed for the genotypes of all the polymorphic sites, and linkage disequilibrium was estimated for the *CCL2* and *CCL5* SNPs. The H-W equilibrium test targeted the allele frequency rates, juxtaposing the values observed with the figures expected, assuming independence between the two, and a Chi-squared distribution had one degree of freedom. The associations of genotypes and haplotypes with the occurrence of endometrial cancer were determined, using a logistic regression model. Differences in genotype and allele prevalence rates were estimated among the various grades and stages of endometrial cancer, using Pearson’s Chi-squared test. Analyses were carried out for multiple SNPs and for haplotypes by the Expectation Maximization (EM) algorithm. First, the initial haplotype frequency values were provided. Passage E consisted then of the recalculation of the expected genotype frequency for the genotypes with haplotypes of uncertainty (in H-W equilibrium), using the haplotype frequency rates. Using the converted genotype frequency rates, the M run calculated each haplotype frequency. Consistent haplotypes were counted for each genotype. Finally, the algorithm converged to the desired haplotype frequency rates. The results were statistically significant at the significance level of *p* ≤ 0.050.

## 3. Results

### 3.1. Study Population

The women with endometrial cancer were significantly older than the non-cancerous control individuals (*p* ≤ 0.001, see Table 1). Diabetes mellitus, arterial hypertension, and endometrial thickness above 5 mm were much more common in the patients with endometrial cancer, compared to those in the control group (*p* ≤ 0.001). According to the FIGO grading, G1, G2, and G3 tumors were determined in 52.14% (61/117), 35.04% (41/117), and 12.82% (15/117) of the women diagnosed with endometrial cancer, respectively. Taking into account the FIGO staging rules, stages I, II, and III were found in 72.2% (83/115), 13.9% (16/115), and 13.9% of the endometrial cancer cases, respectively.

### 3.2. Hardy-Weinberg Equilibrium and Linkage Disequilibrium

The H-W equilibrium was observed for *IL8* −738 T>A and *CCL5* 351 A>G SNPs among all the studied women, as well as for *CXCR2* 1440 G>A and *CCL5* −403 G>A polymorphisms, among the patients with endometrial cancer and non-cancerous individuals, respectively (*p* ≤ 0.050). Regarding *IL8* −845 T>C, the analyzed groups were monomorphic, while H-W was not calculated. In the case of the remaining genotypes, H-W was not preserved (*p* ≤ 0.050). The genotypes within *CCL2* and *CCL5* SNPs were found in the linkage disequilibrium between the cancerous and non-cancerous women (*p* ≤ 0.050).

### 3.3. Genotypes in Chemokine and Chemokine Receptor Gene Polymorphisms

Among the patients with endometrial cancer, TT, TC, and CC genotypes in *CCL2* 903 T>C SNP, were determined in 86.2% (112/130), 3.1% (4/130), and 10.8% (14/130) of the patients, respectively (see Table 4). In the case of *CCL5* −403 G>A polymorphism, GG, GA, and AA variants were found in 56.9% (70/123), 26.8% (33/123), and 16.3% (20/123) of the women, respectively. For *CCL5* 351 A>G SNP, AA, GA, and GG genotypes were observed in 85.0% (108/127), 14.2% (18/127), and 0.8% (1/127) of the patients, respectively. Regarding *IL8* −738 T>A SNP, TT and TA genotypes were found in 99.2% (129/130) and 0.8% (1/130) of the cancerous individuals, respectively. Taking into account the monomorphic *IL8* −845 T>C polymorphism, all the analyzed women carried the TT genotype. In the case of *CXCR2* 1440 G>A SNP, AA, GA, and GG genotypes, were determined in 44.4% (56/126), 35.7% (45/126), and 19.8% (25/126) of the patients, respectively.

Considering the non-cancerous women, TT, CT, and CC genotypes within *CCL2* SNP were observed in 89.3% (133/149), 2.0% (3/149), and 8.7% (13/149) of the patients, respectively. Taking into account *CCL5* −403 G>A polymorphism, GG, GA, and AA genotypes were determined in 66.4% (99/149), 29.5% (44/149), and 4.0% (6/149) of the controls, respectively. In the case of *CCL5* 351 A>G SNP, AA and AG genotypes were found in 83.2% (124/149) and 16.8% (25/149) of the patients, respectively. For *IL8* −738 T>A SNP, TT and TA variants were determined in 99.3% (149/150) and 0.7% (1/150) of the controls, respectively. Regarding *CXCR2* polymorphism, AA, GA, and GG genotypes were observed in 55.7% (83/149), 14.8% (22/149), and 29.5% (44/149) of the patients, respectively.

A single-SNP statistical analysis showed more than a four times higher risk of endometrial cancer in the women with the AA homozygotic status within *CCL5* −403 G>A polymorphism (OR 4.71 95% CI 1.80–12.34 in the codominant model and OR 4.63 95% CI 1.80–11.93 in the recessive model, *p* ≤ 0.050, see Table 4). Similarly, it was observed that the women who were GA heterozygous for the *CXCR2* SNP had an approximately threefold higher risk of endometrial cancer (OR 3.03 95% CI 1.64–5.59 in the codominant model and OR 3.21 95% CI 1.79–5.73 in the overdominant model, *p* ≤ 0.001, see Table 4). The association also remained significant after the correction for age, diabetes mellitus, or arterial hypertension, both in codominant and overdominant models (*p* ≤ 0.050, see Table 5). When adjusted for endometrial thickness above 5 mm, GA heterozygotes in the *CXCR2* polymorphism were associated with about double the risk of cancer in the overdominant model (OR 2.23 95% CI 1.03–4.83, *p* = 0.038).

A multiple-SNP analysis determined A-A haplotypes, for both *CCL5* polymorphisms and T-A-A haplotypes in the range of *CCL2* and *CCL5* SNPs to be associated with an approximately twice higher risk of endometrial cancer (OR 1.84 95% CI 1.21–2.81, and OR 1.71 95% CI 1.10–2.65, respectively, *p* ≤ 0.050, see Table 6 and Table 7).

Taking into account the grades and stages of endometrial cancer, the genotypes for all the analyzed SNPs were similarly distributed among the analyzed groups of cancer (see Appendix A).

### 3.4. Allelic Variants within SNPs of the Chemokine and Chemokine Receptor Genes

Considering the women with endometrial cancer, C and T alleles in *CCL2* polymorphism demonstrated prevalence rates of 12.3% (32/260) and 87.7% (228/260), respectively (see Table 8). In the case of *CCL5* −403 G>A SNP, the prevalence rate of G and A alleles was 70.3% (173/246) and 29.7% (73/246), respectively. For *CCL5* 351 A>G polymorphism, the prevalence rate of A and G alleles was 92.1% (234/254) and 7.9% (20/254), respectively. In the case of *IL8* −738 T>A SNP, T and A alleles revealed prevalence rates of 99.6% (259/260) and 0.4% (1/260), respectively. Regarding *CXCR2* polymorphism, G and A alleles were found with prevalence rates of 37.7% (95/252) and 62.3% (157/252), respectively.

Among the non-cancerous controls, the prevalence rates of C and T alleles in *CCL2* SNP were 9.7% (29/298) and 90.3% (269/298), respectively. Regarding *CCL5* −403 G>A polymorphism, the prevalence rates of G and A alleles were 81.2% (242/298) and 18.8% (56/298), respectively. For *CCL5* 351 A>G SNP, A and G alleles were at the prevalence of 91.6% (273/298) and 8.4% (25/298), respectively. In the case of *IL8* −738 T>A polymorphism, the prevalence rates of T and A alleles were 99.7% (299/300) and 0.3% (1/300), respectively. Taking into account *CXCR2* SNP, the prevalence rates of G and A alleles were 36.9% (110/298) and 63.1% (188/298), respectively.

The differences in the distribution of the G and A alleles within *CCL5* −403 G>A polymorphism were statistically significant when compared between the studied cancerous and non-cancerous groups (*p* = 0.003, see Table 8). For the *CXCR2* SNP, the G and A alleles were differentially distributed among the various grades and stages of endometrial cancer (*p* = 0.043 and *p* = 0.034, respectively, see Appendix A). In turn, all the other polymorphisms demonstrated similar prevalence rates of their alleles among the analyzed patient or cancer groups.

### 3.5. Sample Size Calculation

Given the prevalence rates of the alleles, determined for the polymorphisms analyzed in the reported study, the minimum required sample size should have been 157 women, with a 95% confidence level and a 5% margin of error. The value was obtained relative to the results for *CXCR2* rs1126580.

## 4. Discussion

Our study demonstrated that both the AA homozygotes in *CCL5* rs2107538 and the GA heterozygotes in *CXCR2* rs1126580 had significantly been associated with an increased risk of endometrial cancer. The association of GA genotypes in the *CXCR2* polymorphism with a higher risk of endometrial cancer also remained significant after the adjustments carried out for age, diabetes mellitus, arterial hypertension, and endometrial thickness above 5 mm, which had previously been reported as important risk factors for the cancer studied [26,27,28,29]. Moreover, the G and A alleles in rs1126580 were differentially distributed in various grades and stages of the cancer. An earlier study, which targeted women from the central coast of Tunisia, revealed that *CCL5* rs2107538 had also been correlated with triple-negative breast cancer (TNBC) and hormone receptor-positive BC [30]. Similar to our outcomes, TT genotypes in *CCL5* rs2107538 were associated with an increased risk of BC, and the T allele was more prevalent in the women with the disease when compared to the healthy controls [30]. Moreover, the T allele in rs2107538 was associated with low levels of *CCL5* mRNA in the blood cells of BC patients and correlated, in a dose-dependent manner, with low serum levels of CCL5 among BC and TNBC patients [30]. In turn, a case-control study, conducted on the Southeastern Iranian population, showed that the GA and GA-AA genotypes in rs2107538 had been much more common in women with BC than in healthy subjects [31]. In addition, the A allele in the analyzed polymorphism was significantly more frequently observed in BC patients than in healthy controls, as was the case with the women with endometrial cancer, compared to the control subjects in our study [31]. It was also previously established that *CCL5* rs2107538 was a risk factor for prostate cancer, hepatocellular carcinoma, oral cancer, and papillary thyroid cancer [32,33,34,35]. Regarding the functional significance of *CCL5* rs2107538, this SNP was found to be localized at the binding site for the GATA binding protein 2 (GATA2) and was, thus, suggested to be involved in the regulation of CCL5 transcriptional activity [36]. Rs2107538 has been reported to be possibly associated with an altered binding affinity for GATA2, therefore affecting the *CCL5* mRNA expression levels. An analysis of the GTEx database [37] also showed that the reduced expression levels of *CCL5* mRNA had occurred with the rs2107538 G>A change in whole blood cells [36]. Based on the genetic results, obtained in our project and on the previously reported data on CCL5 levels in endometrial cancer, the A allele in rs2107538 may also contribute to altered *CCL5* expression levels in the disease in question [8,38].

Considering the *CXCR2* rs1126580 polymorphism, its involvement has also been reported in several cancer types, including diffuse large B-cell lymphoma (DLBCL), bile duct cancer, and classic Kaposi sarcoma (CKS) [39,40,41]. A population-based case-control study, conducted in patients with non-Hodgkin lymphoma and in healthy individuals, showed that *CXCR2* rs1126580, as well as *IL1A* rs1800587, *TNF* rs1800629, and *IL4R* rs2107356 polymorphisms, were the strongest predictors of the overall survival (OS) in the patients with DLBCL [41]. In the case of CKS, a lower risk of cancer was found in T-G haplotype carriers for the *CXCR2* rs1126579-rs1126580 polymorphisms [39]. In turn, GA heterozygotes in rs1126580 were associated with an increased risk of periodontitis, although that effect was not observed when adjustments were run for covariates, including age, sex, skin color, and smoking habit [42]. In white non-smokers, the C-T-G/T-C-A haplotype for the *CXCR2* rs2230054-rs1126579-rs1126580 polymorphisms correlated with an increased susceptibility to periodontitis, while the C-T-G/T-C-G haplotype protected against the disease [42]. So far, neither the functional importance of rs1126580 for *CXCR2* expression nor the ability of CXCR2 to bind IL8 has been reported [42,43]. Of note, an increased expression of CXCR2 was observed in endometrial cancer tissues compared to non-cancerous endometrial controls [3,44]. Moreover, the CXCR2 expression was positively associated with the endometrial cancer grade, while being inversely associated with disease-free survival (DFS) [3]. Further research would be justified, regarding the involvement of the rs1126580 polymorphism in the altered CXCR2 expression in endometrial cancer and in the disease prognosis, while its outcomes could bring highly promising results.

In the analyzed cohort of women, we additionally found that the A-A haplotypes for the rs2107538-rs2280789 polymorphisms localized in the *CCL5* gene were associated with an approximately twofold increase in the risk of endometrial cancer. Previously, those two *CCL5* SNPs were also reported to be associated with a higher risk of TNBC [30], as well as having a detrimental effect on the cumulative risk of the acute graft-versus-host disease and on relapse-free survival from human HLA-matched allo-HSCT (allogeneic hematopoietic stem cell transplantation) [30,45]. Similarly, the T allele in *CCL5* rs2107538, and the *CCL5* rs2280789-G, BC risk allele were also correlated with the low levels of *CCL5* mRNA in the blood cells of the BC patients and, in a dose-dependent manner, with the low amounts of CCL5 in the sera of the BC and TNBC patients [30]. It is noteworthy to mention that the CCL5 transcription was previously reported to be regulated mainly by rs2280789, localized in its promoter region, while severely decreasing when the G allele was present [46,47,48]. In African-American, European-American, and combined cohorts, infected with HIV-1, the C allele was also shown to reduce *CCL5* transcription, increasing the progression rate of AIDS [46]. Among the subjects with metastatic colorectal cancer (mCRC), the carriers of any G allele in *CCL5* rs2280789, who had received cetuximab plus FOLFIRI, demonstrated shorter OS when compared to AA homozygotes [47]. Japanese mCRC patients, treated with regorafenib, demonstrated a higher incidence of grade 3+ hand-foot skin reactions, observed in the subjects with GG genotypes for rs2280789 than in the carriers of any A allele [49]. We suggest that, in the case of endometrial cancer, the A allele in rs2280789 may be a significant variant, associated with an increased *CCL5* transcription, possibly resulting in higher CCL5 levels, also previously reported for fibroblast cell populations derived from human endometrial cancer tissues [38].

A further multiple-SNP analysis in our study also showed that the T-A-A haplotypes for the *CCL2* rs4586-*CCL5* rs2107538-*CCL5* rs2280789 polymorphisms had correlated with an increased risk of endometrial cancer. *CCL2* rs4586 was previously shown to be involved in several other cancer types, including CRC, locoregional gastric cancer (LRGC), and BC [50,51,52,53]. Regarding the KRAS wild-type mCRC patients, treated with FOLFIRI/bevacizumab, *CCL2* rs4586 was identified as a major OS predictor [51]. In turn, KRAS mutant tumors of the mCRC patients with C alleles of both *CCL2* rs4586 and *IRF3* rs2304205 were associated with better PFS [54]. Among Korean patients with CRC, the T allele in rs4586 correlated with favorable OS, observed in both univariate and multivariate analyses [53]. Taking into account LRGC, US patients with at least one T allele in *CCL2* rs4586 demonstrated significantly shorter OS than CC homozygotes, while longer OS was observed in the Japanese cohort [52].

In premenopausal European American women, *CCL2* rs4586 was also correlated with an increased risk of BC, although the effect disappeared when the analysis was conducted on a larger cohort of participants in the Women’s Circle of Health Study [50]. Given the functional importance of *CCL2* rs4586, an analysis, performed using the GTEx database, showed that the SNP was an expression quantitative trait locus (eQTL) and correlated with higher levels of *CCL13* mRNA in various human tissues [55]. In the Chinese population, the minor T allele in *CCL2* rs4586 was found to be associated with significantly elevated serum CCL2 levels, which also correlated with a higher risk of chronic obstructive pulmonary disease [56]. On the other hand, a case-control study, conducted on a Chinese Han population, showed that the CC genotype in rs4586 had been associated with elevated plasma CCL2 levels compared to the TT genotype [57]. An ethnicity-dependent function of the rs4586 polymorphism was then suggested [57]. In endometrial cancer, the T allele in *CCL2* rs4586 may also be involved in the elevated CCL2 production previously established for this type of malignancy [58,59].

## 5. Conclusions

*CCL2* rs4586, *CCL5* rs2107538 and rs2280789, as well as *CXCR2* rs1126580, seem to be significantly associated with an increased risk of endometrial cancer in the population of Polish women. It may also be suggested that the T allele in *CCL2* rs4586 and the A allele in *CCL5* rs2280789 are likely involved in the increased levels of CCL2 and CCL5, respectively. However, further research projects would be beneficial to unveil in detail the role of the reported polymorphisms in the altered function of the encoded molecules and signaling pathways affected in endometrial cancer. The presented results may also be useful for the development of new therapeutic strategies in endometrial cancer, based on molecular/genetic profiling.

## Figures and Tables

**Figure 1 cancers-15-05416-f001:**
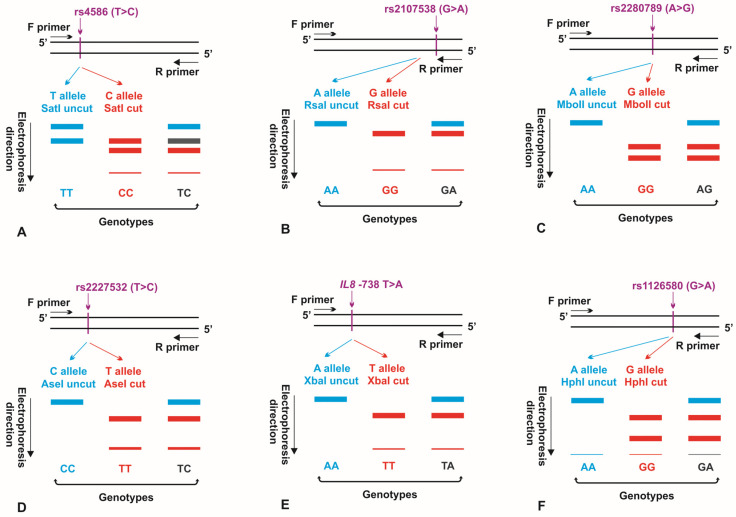
Schematic representation of PCR-RFLP assays, used to determine genotypes for the *CCL2* rs4586 (**A**), *CCL5* rs2107538 (**B**), *CCL5* rs2280789 (**C**), *IL8* rs2227532 (**D**), *IL8* −738 T>A (**E**), and *CXCR2* rs1126580 (**F**) polymorphisms. The alleles of the polymorphic sites, not recognized by appropriate restriction enzymes, as well as DNA fragments and genotypes, obtained when no restriction digestion occurred in the polymorphism, are marked in blue. The alleles of the polymorphic regions, recognized by the endonucleases used in the research, as well as DNA fragments and genotypes, obtained as result of restriction digestion within the polymorphism, are shown in red. F primer: forward primer; R primer: reverse primer.

**Figure 2 cancers-15-05416-f002:**
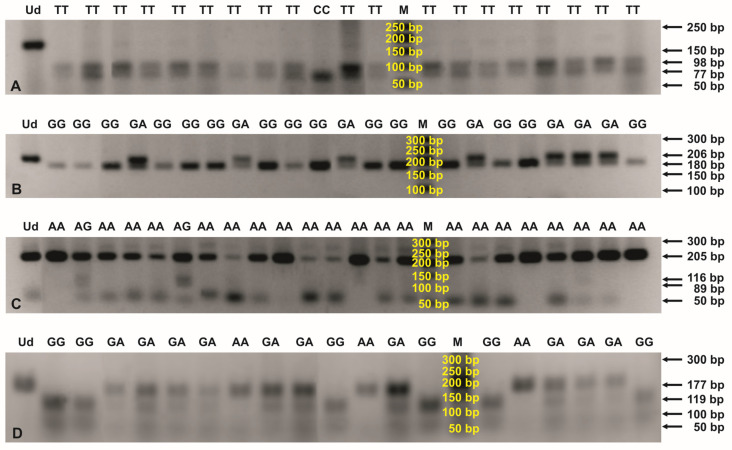
PCR-RFLP profiles for genotyping the *CCL2* rs4586 (**A**), *CCL5* rs2107538 (**B**), *CCL5* rs2280789 (**C**), and *CXCR2* rs1126580 (**D**) polymorphisms. Digestions of the PCR products were performed with SatI (**A**), RsaI (**B**), MboII (**C**), and HphI (**D**) endonucleases and then separated in 2% agarose gels, stained with ethidium bromide. The numbers to the right of the electropherograms show the lengths of the separated DNA fragments. M: 50 bp DNA marker; Ud: undigested PCR product; AA, AG, CC, GA, GG, and TT: genotypes in the tested SNPs.

**Table 1 cancers-15-05416-t001:** Characteristic features of women with endometrial cancer and of non-cancerous controls.

Characteristic Features	Females with Endometrial Cancer	Non-Cancerous Controls	*p*-Value ^a^
Examined women; n ^b^ (%)	132 (46.8)	150 (53.2)	
Age (years)	61 (44–88)	50 (36–75)	≤0.001
Diabetes mellitus (%)	23/132 (17.4)	2/140 (1.4)	≤0.001
Arterial hypertension (%)	71/132 (53.8)	22/140 (15.7)	≤0.001
FIGO ^c^ grade			
G1 (%)	61/117 (52.14)		
G2 (%)	41/117 (35.04)		
G3 (%)	15/117 (12.82)		
FIGO stage			
I (%)	83/115 (72.2)		
II (%)	16/115 (13.9)		
III (%)	16/115 (13.9)		
Endometrial thickness (mm)	6.5 (1.2–116.0)	11 (1.5–14.0)	≤0.001
>5 mm (%)	97/116 (83.6)	59/95 (62.1)	≤0.001

^a^ *p* ≤ 0.050 is considered significant; ^b^ n, the number of examined women; ^c^ FIGO, International Federation of Gynecology and Obstetrics. Continuous variables are presented as medians (minimum–maximum) and categorical as numbers (%).

**Table 2 cancers-15-05416-t002:** Oligonucleotides, annealing temperatures, and amplicon lengths in PCR assays for *CCL2*, *CCL5*, *IL8*, and *CXCR2* polymorphisms.

Gene	GenBank Accession No. ^a^	SNP ^b^ Name	Oligonucleotides (5′-3′)		Annealing Temperature (°C)	Amplicon Length (bps) ^c^
*CCL2*	NC_000017.11	903 T>C	External	For: TCCTCATGACTCTTTTCTGC	52	263
		(rs4586)		Rev: GAGGCTTGTCCCTTGCTCCAC		
			Internal	For: TTCCAGATGCAATCAATGCC	55	173
				Rev: TCCACAAGGAAGAACTTCAG		
*CCL5*	NC_000017.11	−403 G>A		For: CACAAGAGGACTCATTCCAACTCA	55	206
		(rs2107538)		Rev: GTTCCTGCTTATTCATTACAGATCGTA		
		351 A>G	External	For: CCTGGTCTTGACCACCACA	52	343
		(rs2280789)		Rev: GCTGACAGGCATGAGTCAGA		
			Internal	For: AGCATGAATACTGTCTGAACCC	55	205
				Rev: GCTACTCATGACTACTAATCTCC		
*IL8*	NC_000004.12	−845 T>C	External	For: AACCCAGCAGCTCCAGTG	52	534
		(rs2227532)		Rev: AGATAAGCCAGCCAATCATT		
		−738 T>A	Internal	For: CAACTGCCTTTGGAAGATTCTG	55	185
				Rev: GGAGTAACTTTCTGAGTAATGTGG		
*CXCR2*	NC_000002.12	1440 G>A	External	For: CCCCATTGTGGTCACAGGACG	52	343
		(rs1126580)		Rev: GCCTCCCAAGTAGCTGTGATTA		
			Internal	For: GGCACTCTATGTTCTAAGAAGTG	56.4	181
				Rev: GATTACAGGCACTCACCACC		

^a^ No., number; ^b^ SNP, single nucleotide polymorphism; ^c^ bps, base pairs.

**Table 3 cancers-15-05416-t003:** Endonucleases and restriction profiles in PCR-RFLP assays.

Polymorphism	Endonuclease	Profile (bps) ^a^
*CCL2*	SatI	CC: 77, 66, 32
903 T>C		TC: 98, 77, 66, 32
		TT: 98, 77
*CCL5*	RsaI	GG: 180, 26
−403 G>A		GA: 206, 180, 26
		AA: 206
*CCL5*	MboII	AA: 205
351 A>G		AG: 205, 116, 89
		GG: 116, 89
*IL8*	AseI	TT: 142, 43
−845 T>C		TC: 185, 142, 43
		CC: 185
*IL8*	XbaI	TT: 152, 33
−738 T>A		TA: 185, 152, 33
		AA: 185
*CXCR2*	HphI	GG: 119, 58, 4
1440 G>A		GA: 177, 119, 58, 4
		AA: 177, 4

^a^ bps, base pairs.

**Table 4 cancers-15-05416-t004:** Relationship between genotypes in *CCL2*, *CCL5*, *IL8*, and *CXCR2* polymorphisms and the occurrence of endometrial cancer.

Gene Polymorphism	Genetic Model	Genotype	Genotype Prevalence Rates; n (%) ^a^		OR ^b^ (95% CI) ^c^	*p*-Value ^d^
			Endometrial Cancer	Non-Cancerous Controls		
*CCL2*	Codominant	T/T	112 (86.2%)	133 (89.3%)	1.00	
903 T>C		T/C	4 (3.1%)	3 (2.0%)	1.58 (0.35–7.22)	
		C/C	14 (10.8%)	13 (8.7%)	1.28 (0.58–2.83)	0.710
	Dominant	T/T	112 (86.2%)	133 (89.3%)	1.00	
		T/C-C/C	18 (13.8%)	16 (10.7%)	1.34 (0.65–2.74)	0.430
	Recessive	T/T-T/C	116 (89.2%)	136 (91.3%)	1.00	
		C/C	14 (10.8%)	13 (8.7%)	1.26 (0.57–2.79)	0.560
	Overdominant	T/T-C/C	126 (96.9%)	146 (98.0%)	1.00	
		T/C	4 (3.1%)	3 (2.0%)	1.54 (0.34–7.03)	0.570
*CCL5*	Codominant	G/G	70 (56.9%)	99 (66.4%)	1.00	
−403 G>A		G/A	33 (26.8%)	44 (29.5%)	1.06 (0.61–1.83)	
		A/A	20 (16.3%)	6 (4.0%)	4.71 (1.80–12.34)	0.002
	Dominant	G/G	70 (56.9%)	99 (66.4%)	1.00	
		G/A-A/A	53 (43.1%)	50 (33.6%)	1.50 (0.92–2.45)	0.110
	Recessive	G/G-G/A	103 (83.7%)	143 (96.0%)	1.00	
		A/A	20 (16.3%)	6 (4.0%)	4.63 (1.80–11.93)	≤0.001
	Overdominant	G/G-A/A	90 (73.2%)	105 (70.5%)	1.00	
		G/A	33 (26.8%)	44 (29.5%)	0.88 (0.51–1.49)	0.620
*CCL5*	Codominant	A/A	108 (85.0%)	124 (83.2%)	1.00	
351 A>G		A/G	18 (14.2%)	25 (16.8%)	0.83 (0.43–1.60)	
		G/G	1 (0.8%)	0 (0.0%)	NA (0.00–NA) ^e^	0.390
	Dominant	A/A	108 (85.0%)	124 (83.2%)	1.00	
		A/G-G/G	19 (15.0%)	25 (16.8%)	0.87 (0.46–1.67)	0.680
	Recessive	A/A-A/G	126 (99.2%)	149 (100.0%)	1.00	
		G/G	1 (0.8%)	0 (0.0%)	NA (0.00–NA)	0.210
	Overdominant	A/A-G/G	109 (85.8%)	124 (83.2%)	1.00	
		A/G	18 (14.2%)	25 (16.8%)	0.82 (0.42–1.58)	0.550
*IL8*	-	T/T	129 (99.2%)	149 (99.3%)	1.00	
−738 T>A		T/A	1 (0.8%)	1 (0.7%)	1.16 (0.07–18.65)	0.920
*CXCR2*	Codominant	A/A	56 (44.4%)	83 (55.7%)	1.00	
1440 G>A		G/A	45 (35.7%)	22 (14.8%)	3.03 (1.64–5.59)	
		G/G	25 (19.8%)	44 (29.5%)	0.84 (0.46–1.53)	≤0.001
	Dominant	A/A	56 (44.4%)	83 (55.7%)	1.00	0.062
		G/A-G/G	70 (55.6%)	66 (44.3%)	1.57 (0.98–2.53)	
	Recessive	A/A-G/A	101 (80.2%)	105 (70.5%)	1.00	0.063
		G/G	25 (19.8%)	44 (29.5%)	0.59 (0.34–1.04)	
	Overdominant	A/A-G/G	81 (64.3%)	127 (85.2%)	1.00	≤0.001
		G/A	45 (35.7%)	22 (14.8%)	3.21 (1.79–5.73)	

^a^ n, the number of examined women; ^b^ OR, odds ratio; ^c^ 95% CI, confidence interval; ^d^ logistic regression model; *p* ≤ 0.050 is considered significant; ^e^ NA, not analyzed.

**Table 5 cancers-15-05416-t005:** Associations of *CXCR2* 1440 G>A polymorphism with endometrial cancer adjusted by selected covariates.

Covariate	Genetic Model	Genotype	Genotype Prevalence, n ^a^ (%)		OR ^b^ (95% CI ^c^)	*p*-Value ^d^
			Endometrial Cancer	Non-Cancerous Controls		
Age	Codominant	A/A	75 (59.5%)	52 (50.0%)	1.00	0.020
		G/A	14 (11.1%)	31 (29.8%)	3.62 (1.42–9.20)	
		G/G	37 (29.4%)	21 (20.2%)	1.15 (0.51–2.57)	
	Dominant	A/A	75 (59.5%)	52 (50.0%)	1.00	0.082
		G/A-G/G	51 (40.5%)	52 (50.0%)	1.82 (0.92–3.60)	
	Recessive	A/A-G/A	89 (70.6%)	83 (79.8%)	1.00	0.630
		G/G	37 (29.4%)	21 (20.2%)	0.83 (0.39–1.78)	
	Overdominant	A/A-G/G	112 (88.9%)	73 (70.2%)	1.00	0.005
		G/A	14 (11.1%)	31 (29.8%)	3.46 (1.42–8.43)	
Diabetes mellitus	Codominant	A/A	69 (58.0%)	52 (50.0%)	1.00	0.002
		G/A	14 (11.8%)	31 (29.8%)	3.00 (1.42–6.32)	
		G/G	36 (30.2%)	21 (20.2%)	0.71 (0.36–1.41)	
	Dominant	A/A	69 (58.0%)	52 (50.0%)	1.00	0.290
		G/A-G/G	50 (42.0%)	52 (50.0%)	1.35 (0.78–2.34)	
	Recessive	A/A-G/A	83 (69.8%)	83 (79.8%)	1.00	0.056
		G/G	36 (30.2%)	21 (20.2%)	0.53 (0.28–1.03)	
	Overdominant	A/A-G/G	105 (88.2%)	73 (70.2%)	1.00	0.001
		G/A	14 (11.8%)	31 (29.8%)	3.33 (1.62–6.82)	
Arterial hypertension	Codominant	A/A	69 (58.0%)	52 (50.0%)	1.00	0.015
		G/A	14 (11.8%)	31 (29.8%)	2.50 (1.13–5.52)	
		G/G	36 (30.2%)	21 (20.2%)	0.69 (0.34–1.41)	
	Dominant	A/A	69 (58.0%)	52 (50.0%)	1.00	0.540
		G/A-G/G	50 (42.0%)	52 (50.0%)	1.20 (0.67–2.16)	
	Recessive	A/A-G/A	83 (69.8%)	83 (79.8%)	1.00	0.077
		G/G	36 (30.2%)	21 (20.2%)	0.54 (0.27–1.08)	
	Overdominant	A/A-G/G	105 (88.2%)	73 (70.2%)	1.00	0.007
		G/A	14 (11.8%)	31 (29.8%)	2.81 (1.31–6.01)	
Endometrial thickness >5 mm	Codominant	A/A	42 (51.2%)	43 (47.2%)	1.00	0.075
		G/A	12 (14.6%)	28 (30.8%)	1.96 (0.87–4.45)	
		G/G	28 (34.1%)	20 (22.0%)	0.70 (0.34–1.46)	
	Dominant	A/A	42 (51.2%)	43 (47.2%)	1.00	0.760
		G/A-G/G	40 (48.8%)	48 (52.8%)	1.10 (0.60–2.03)	
	Recessive	A/A-G/A	54 (65.8%)	71 (78.0%)	1.00	0.120
		G/G	28 (34.1%)	20 (22.0%)	0.57 (0.29–1.15)	
	Overdominant	A/A-G/G	70 (85.4%)	63 (69.2%)	1.00	0.038
		G/A	12 (14.6%)	28 (30.8%)	2.23 (1.03–4.83)	

^a^ n, the number of examined women; ^b^ OR, odds ratio; ^c^ 95% CI, confidence interval; ^d^ logistic regression model; *p* ≤ 0.050 is considered significant.

**Table 6 cancers-15-05416-t006:** Relationships between the haplotypes for *CCL5* SNPs and the occurrence of endometrial cancer.

Alleles in *CCL5* SNPs ^a^		Haplotype Prevalence		OR ^b^ (95% CI ^c^)	*p*-Value ^d^
−403 G>A	351 A>G	Endometrial Cancer	Non-Cancerous Controls		
G	A	0.696	0.801	1.00	
A	A	0.225	0.116	1.84 (1.21–2.81)	0.005
A	G	0.068	0.072	1.09 (0.55–2.15)	0.800
G	G	0.010	0.012	0.93 (0.15–5.77)	0.940

^a^ SNPs, single nucleotide polymorphisms; ^b^ OR, odds ratio; ^c^ 95% CI, confidence interval; ^d^ logistic regression model; *p* ≤ 0.050 is considered significant; global haplotype association *p*-value was determined to be 0.033.

**Table 7 cancers-15-05416-t007:** Associations of the haplotypes for *CCL2* and *CCL5* polymorphisms with the onset of endometrial cancer.

Alleles in *CCL2* and *CCL5* SNPs ^a^			Haplotype Prevalence		OR ^b^ (95% CI ^c^)	*p*-Value ^d^
*CCL2* 903 T>C	*CCL5*		Endometrial Cancer	Non-Cancerous Controls		
	−403 G>A	351 A>G				
T	G	A	0.613	0.720	1.00	
T	A	A	0.190	0.108	1.71 (1.10–2.65)	0.017
C	G	A	0.084	0.081	1.06 (0.64–1.77)	0.820
T	A	G	0.068	0.063	1.24 (0.62–2.51)	0.540
C	A	A	0.036	0.008	4.10 (0.90–18.66)	0.069

^a^ SNPs, single nucleotide polymorphisms; ^b^ OR, odds ratio; ^c^ 95% CI, confidence interval; ^d^ logistic regression model; *p* ≤ 0.050 is considered significant; global haplotype association *p*-value was determined to be 0.046.

**Table 8 cancers-15-05416-t008:** Distribution of the alleles, localized within *CCL2*, *CCL5*, *IL8*, and *CXCR2* polymorphisms.

Gene Polymorphism	Allele	No. ^a^ of Carriers with Specific Alleles (%)		*p*-Value ^b^
		Endometrial Cancer	Non-Cancerous Controls	
*CCL2*	T	228 (87.7%)	269 (90.3%)	0.331
*903 T>C*	C	32 (12.3%)	29 (9.7%)	
*CCL5*	G	173 (70.3%)	242 (81.2%)	0.003
*−403 G>A*	A	73 (29.7%)	56 (18.8%)	
*CCL5*	A	234 (92.1%)	273 (91.6%)	0.825
*351 A>G*	G	20 (7.9%)	25 (8.4%)	
*IL8*	T	259 (99.6%)	299 (99.7%)	0.919
*−738 T>A*	A	1 (0.4%)	1 (0.3%)	
*CXCR2*	G	95 (37.7%)	110 (36.9%)	0.849
*1440 G>A*	A	157 (62.3%)	188 (63.1%)	

^a^ No., number; ^b^ Pearson’s Chi-squared test; *p* ≤ 0.050 is considered significant.

## Data Availability

The data presented in this study are available on request from the corresponding author.

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
