# Peer review of "Associations between Single Nucleotide Polymorphisms from the Genes of Chemokines and the CXCR2 Chemokine Receptor and an Increased Risk of Endometrial Cancer"

_cancers, 2023, doi:10.3390/cancers15225416_

Round 1
Reviewer 1 Report
Comments and Suggestions for Authors
I read with great interest the Manuscript titled " Associations between single nucleotide polymorphisms from genes of chemokines and the CXCR2 chemokine receptor and the increased risk of endometrial cancer”, topic interesting enough to attract readers' attention.
Although the manuscript can be considered already of good quality, I would suggest following recommendations:
- I suggest a round of language revision, in order to correct few typos and improve readability.
- In recent years the classification of endometrial cancer has evolved significantly. Considering results and topics of this study, I suggest that authors to add references to discuss recent evidence about molecular mechanisms and the modulation of the treatment of endometrial cancer. I would be glad if the authors discuss this important point, referring to PMID: 36983243 and 36979434.
Because of these reasons, the article should be revised and completed. Considering all these points, I think it could be of interest to the readers and, in my opinion, it deserves the priority to be published after minor revisions
Comments on the Quality of English Language
I suggest a round of language revision, in order to correct few typos and improve readability
Author Response
Reviewer’s comment:
- I suggest a round of language revision, in order to correct few typos and improve readability.
Response:
Dear Reviewer, thank you very much for your suggestion. Following your advice, the manuscript has been linguistically revised again to correct any typos and improve readability.
Reviewer’s comment:
- In recent years the classification of endometrial cancer has evolved significantly. Considering results and topics of this study, I suggest that authors to add references to discuss recent evidence about molecular mechanisms and the modulation of the treatment of endometrial cancer. I would be glad if the authors discuss this important point, referring to PMID: 36983243 and 36979434.
Response:
In accordance with the Reviewer's important note, we have improved the Introduction section of the manuscript with the recommended references PMID: 36983243 and 36979434 regarding a new approach to the classification of endometrial cancer and the molecular mechanisms and modulation of the treatment of this disease. In the Conclusions section, we have also added an important new sentence containing an innovative perspective on endometrial cancer treatment methods.
Reviewer 2 Report
Comments and Suggestions for Authors
This research analyzed several cytokines and cytokines receptors in endometrial cancer using the PCR-restriction fragment length polymorphism (RFLP) technique in a series of 132 females with endometrial cancer and 150 non-cancerous control. Sanger was also used to confirm the SNP analysis (although no image is shown). The manuscript is well written, and it is easy to understand. The authors concluded that association of CCL2, CCL5, and CXCR2 with risk of endometrial cancer was found. Nevertheless, the pathogenic mechanism is not explored and it is out of the scope of this study. To improve the manuscript, the authors may address the following comments:
Comments:
(1) Please add the catalog number of the Syngen Biotech reagent.
(2) Please add the webpage link of the FIGO criteria.
(3) Please add "IRB" before the "the approval number").
(4) The study included 132 cancer and 150 controls. Is the design a "matched-pair analysis"? Age, DM, hypertension, endometrial thickness variables were all statistically different. Therefore, test and controls are different?
(5) Figure 1 shows results of PCR-RFLP. The ladder M has the 50bp DNA marker. May you please add the bp of the marks of the ladder using for example yellow fonts?
(6) Could you please provide description of the "Expectation Maximization (EM) algorithm"?
(7) In molecular biology, restriction fragment length polymorphism (RFLP) is a technique that exploits variations in homologous DNA sequences, known as polymorphisms, populations, or species or to pinpoint the locations of genes within a sequence. RFLP analysis is now largely obsolete due to the emergence of inexpensive DNA sequencing technologies. What is the difference between RFLP and PCR-RFLP? Could you please make a figure showing the schematic representation of the PCR-RFLP technique that has been used in this article? With a nice figure, the analysis can be easily understood.
(8) The PCR products were sequenced by Sanger. All of them, or ony a selection? Only CCL2 and CCL5?
(9) Could you please provide description of the Hardy-Weinberg (H-W) equilibrium analysis?
(10) Since in tables 4 to 8 multiples comparison are being made. Is any kind of multiple comparisons correction necessary? For example, instead of cutoff of 0.05, would it be more appropriate to use 0.01?
(11) Sorry if I understood wrongly, but, why in Table 7 both CCL2 and CCL5 are included in the analysis? Table 6 was CCL5. Should Table 7 be only CCL2?
(12) Regarding this sentence in the abstract "AA homozygotes in CCL5 rs2107538 were associated with more than a quadruple risk of endometrial cancer (P≤0.050). GA heterozygotes in the CXCR2 SNP were correlated with approximately threefold higher cancer risk (P≤0.001)." Could you please highlight the lines in the results sections that contain this result?
(13) Could you please also highlight in the results section the sentences that show the result for this sentence of the abstract ("A haplotypes for the CCL2 and CCL5 SNPs were associated with about a twofold risk of endometrial cancer (P≤0.050)" ) ?
(14) Does endometrial cancer express high RNA and protein levels of CCL5 and CXCR2? Are this two markers associated with prognosis?
Author Response
Reviewer’s comment:
(1) Please add the catalog number of the Syngen Biotech reagent.
Response:
Dear Reviewer, we have added the catalog number of the Syngen Biotech reagent.
Reviewer’s comment:
(2) Please add the webpage link of the FIGO criteria.
Response:
Dear Reviewer, we have added the webpage link to the FIGO criteria.
Reviewer’s comment:
(3) Please add "IRB" before the "the approval number").
Response:
Dear Reviewer, we have added "IRB" before "the approval number".
Reviewer’s comment:
(4) The study included 132 cancer and 150 controls. Is the design a "matched-pair analysis"? Age, DM, hypertension, endometrial thickness variables were all statistically different. Therefore, test and controls are different?
Response:
Dear Reviewer, in the presented case-control genetic association study, we did not perform matched-pair analysis. Due to differences in age, diabetes, hypertension, and endometrial thickness between the study and control groups of women, we adjusted the results for the association of genotypes with endometrial cancer using these variables. The presented results showed that the study and control groups differed in age, diabetes, hypertension and endometrial thickness. However, the aim of the study was to compare the distribution of genetic changes in selected SNPs between women with endometrial cancer and women from the control group without cancer.
Reviewer’s comment:
(5) Figure 1 shows results of PCR-RFLP. The ladder M has the 50bp DNA marker. May you please add the bp of the marks of the ladder using for example yellow fonts?
Response:
Dear Reviewer, following your advice, we have added bp ladder characters using a yellow font.
Reviewer’s comment:
(6) Could you please provide description of the "Expectation Maximization (EM) algorithm"?
Response:
Dear Reviewer, in accordance with your note, we have included a description of the Expectation Maximization (EM) algorithm.
Reviewer’s comment:
(7) In molecular biology, restriction fragment length polymorphism (RFLP) is a technique that exploits variations in homologous DNA sequences, known as polymorphisms, populations, or species or to pinpoint the locations of genes within a sequence. RFLP analysis is now largely obsolete due to the emergence of inexpensive DNA sequencing technologies. What is the difference between RFLP and PCR-RFLP? Could you please make a figure showing the schematic representation of the PCR-RFLP technique that has been used in this article? With a nice figure, the analysis can be easily understood.
Response:
Dear Reviewer, restriction fragment length polymorphism (RFLP) and PCR-restriction fragment length polymorphism (PCR-RFLP) are two different techniques used to detect differences in homologous DNA sequences. RFLP is a molecular marker that is specific to a single clone/restriction enzyme combination. Most RFLP markers are co-dominant and highly locus-specific. An RFLP probe is a labeled DNA sequence that hybridizes with one or more fragments of the digested DNA sample after they were separated by gel electrophoresis, thus revealing a unique blotting pattern characteristic to a specific genotype at a specific locus. PCR-RFLP is a modification of RFLP that uses PCR to amplify very small amounts of DNA to the levels required for RFLP analysis. Therefore, PCR-RFLP is faster and more efficient than RFLP. However, PCR-RFLP requires a higher level of technical expertise and is more prone to errors than RFLP. In conclusion, both RFLP and PCR-RFLP are viable techniques for detecting differences in homologous DNA sequences. However, the choice between the two depends on the specific requirements of the experiment and the available resources.
Taking into account the Reviewer's suggestion, we have prepared additional Figure 1 showing a schematic representation of the PCR-RFLP technique used in our article. Therefore, the analysis is easy to understand.
Reviewer’s comment:
(8) The PCR products were sequenced by Sanger. All of them, or only a selection? Only CCL2 and CCL5?
Response:
Dear Reviewer, in section “2.2. Genotypes within SNPs of Chemokine and Chemokine Receptor Genes”, we wrote earlier: “The selected PCR products were additionally sequenced by the Sanger method at the Genomed Joint-Stock Company (Warsaw, Poland), to corroborate the genotypes previously estimated by the PCR-RFLP method. For CCL2 SNP, DNA fragments were sequenced for ten TT, six CT, and six CC genotypes. In case of CCL5 -403 G>A polymorphism, nine GG and 12 GA genotypes were verified by sequencing. Regarding CCL5 351 A>G SNP, PCR products were sequenced for seven AA, and eight AG genotypes. In case of IL8 -845 T>C SNP, sequencing was performed for 14 TT genotypes, while in case of IL8 -738 polymorphism – for seven TT and two TA genotypes.”
Reviewer’s comment:
(9) Could you please provide description of the Hardy-Weinberg (H-W) equilibrium analysis?
Response:
Dear Reviewer, taking into account your advice, we have included a description of the Hardy-Weinberg (H-W) equilibrium analysis.
Reviewer’s comment:
(10) Since in tables 4 to 8 multiples comparison are being made. Is any kind of multiple comparisons correction necessary? For example, instead of cutoff of 0.05, would it be more appropriate to use 0.01?
Response:
Dear Reviewer, of the tables 4 to 8, only Table 5 presents the adjusted results of the statistical analysis. We performed an adjusted analysis to exclude the potential influence of selected characteristics of the study women on the occurrence and development of endometrial cancer.
The P-value is a statistical measure that helps determine the probability of observing a result as extreme as the one obtained, assuming that the null hypothesis is true. It is used to test the significance of a hypothesis test and is usually compared to a significance level, which is often set to 0.05 or 0.01. A significance level of 0.05 means that if the P-value is less than or equal to 0.05, the null hypothesis can be rejected with 95% confidence. Similarly, a significance level of 0.01 means that if the P-value is less than or equal to 0.01, the null hypothesis can be rejected with 99% confidence. The smaller the P-value, the more likely you are to reject the null hypothesis. However, it is important to note that a small P-value does not necessarily mean that the alternative hypothesis is true. It only means that the observed data is unlikely to have occurred by chance if the null hypothesis were true.
We considered the results to be statistically significant at the significance level of P≤0.050, in accordance with the results obtained using the SNPStats software.
Reviewer’s comment:
(11) Sorry if I understood wrongly, but, why in Table 7 both CCL2 and CCL5 are included in the analysis? Table 6 was CCL5. Should Table 7 be only CCL2?
Response:
Dear Reviewer, in Table 7, both CCL2 and CCL5 polymorphisms were included in the analysis because we wanted to test possible associations of the haplotypes for these SNPs with the onset of endometrial cancer. In Table 6 we presented the results of possible haplotype associations for CCL5 SNPs. In the case of CCL2, in the present study we only tested the rs4586 polymorphism (903 T>C). Therefore, the genotype analysis for CCL2 rs4586 is presented in Table 4.
Reviewer’s comment:
(12) Regarding this sentence in the abstract "AA homozygotes in CCL5 rs2107538 were associated with more than a quadruple risk of endometrial cancer (P≤0.050). GA heterozygotes in the CXCR2 SNP were correlated with approximately threefold higher cancer risk (P≤0.001)." Could you please highlight the lines in the results sections that contain this result?
Response:
Dear Reviewer, in accordance with your note, in the Results section we have highlighted the lines containing the results "AA homozygotes in CCL5 rs2107538 were associated with more than a quadruple risk of endometrial cancer (P≤0.050). GA heterozygotes in the CXCR2 SNP were correlated with approximately threefold higher cancer risk (P≤0.001)."
Reviewer’s comment:
(13) Could you please also highlight in the results section the sentences that show the result for this sentence of the abstract ("A haplotypes for the CCL2 and CCL5 SNPs were associated with about a twofold risk of endometrial cancer (P≤0.050)")?
Response:
Dear Reviewer, in accordance with your note, in the Results section we have highlighted the lines containing the results "A haplotypes for the CCL2 and CCL5 SNPs were associated with about a twofold risk of endometrial cancer (P≤0.050)".
Reviewer’s comment:
(14) Does endometrial cancer express high RNA and protein levels of CCL5 and CXCR2? Are these two markers associated with prognosis?
Response:
Dear Reviewer, increased expression of both CCL5 and CXCR2 has been reported in endometrial cancer tissues compared to non-cancerous endometrial controls. Moreover, CXCR2 expression was positively associated with endometrial cancer grade, while inversely with disease-free survival (DFS). No associations have been reported between CCL5 expression and prognosis in endometrial cancer. We have added the above CXCR2 data to the Discussion section.